# Sputum Galactomannan Has Utility in the Diagnosis of Chronic Pulmonary Aspergillosis

**DOI:** 10.3390/jof8020188

**Published:** 2022-02-14

**Authors:** Ali Nuh, Newara Ramadan, Anand Shah, Darius Armstrong-James

**Affiliations:** 1Laboratory Medicine, Department of Microbiology, Royal Brompton Hospital, Guy’s and St Thomas’ NHS Foundation Trust, London SW3 6NP, UK; n.ramadan@rbht.nhs.uk (N.R.); d.armstrong@imperial.ac.uk (D.A.-J.); 2Department of Respiratory Medicine, Royal Brompton Hospital, Guy’s and St Thomas’ NHS Foundation Trust, London SW3 6NP, UK; a.shah2@rbht.nhs.uk; 3MRC Centre of Global Infectious Disease Analysis, Department of Infectious Disease Epidemiology, School of Public Health, Imperial College, London SW7 2AZ, UK; 4MRC Centre for Molecular Bacteriology and Infection, Department of Infectious Diseases, Imperial College London, London SW7 2BX, UK

**Keywords:** *Aspergillus species*, pulmonary aspergillosis, galactomannan, sputum, diagnosis

## Abstract

Diagnosis of pulmonary aspergillosis (PA), a fungal disease caused by *Aspergillus species,* is challenging since symptoms are unspecific. The galactomannan (antigen secreted by *Aspergillus species*) test in bronchoalveolar lavage (BAL) fluid is a valuable diagnostic adjunct test in the diagnosis of PA. However, BAL collection is invasive and may not be suitable to severely ill patients. Sputum is non-invasive, easily collected, and lung specific and may be an alternative to BAL. The aim of this research was to retrospectively evaluate the utility of sputum galactomannan in the diagnosis of pulmonary aspergillosis in patients with chronic respiratory diseases and to estimate the sputum galactomannan cut-off value. We collected data from patients with clinical suspicion of pulmonary aspergillosis who had sputum galactomannan, culture, and *Aspergillus* IgG tests performed within four weeks. Sputum galactomannan was validated against the clinical diagnosis of aspergillosis, *Aspergillus* culture, and *Aspergillus* IgG tests. In total, 218 patients met inclusion criteria. Overall, sputum GM showed satisfactory agreement with clinical diagnosis of aspergillosis, *Aspergillus* culture, and *Aspergillus* IgG. When a receiver operating characteristic curve was constructed using *Aspergillus* culture/IgG and clinical diagnosis, the same cut-off (CO) of 0.71 (AUC: 0.83; CI: 0.69–0.86, *p* < 0.001) was determined. Against clinical diagnosis, sputum GM gave sensitivity and specificity of 70% and 71%, respectively. Sensitivity of 77% and specificity of 78% were found when sputum GM was evaluated against *Aspergillus* culture/IgG. In conclusion, this study showed that sputum galactomannan antigen testing has utility in the diagnosis of chronic forms of pulmonary aspergillosis and further prospective validation is indicated.

## 1. Introduction

Aspergillosis refers to a spectrum of fungal diseases which are caused by *Aspergillus species*; these include invasive aspergillosis (IA), allergic bronchopulmonary aspergillosis (ABPA), and chronic pulmonary aspergillosis (CPA). IA is a fatal and difficult to treat infection that mainly affects immunocompromised and critically ill patients, while CPA manifests primarily in patients with chronic respiratory disease. ABPA is a hypersensitive reaction to *Aspergillus species* and is common in patient with asthma and cystic fibrosis [1,2]. Diagnosis of aspergillosis is challenging due to unspecific symptoms and is often achieved by combination of clinical, radiological, and laboratory tests. Laboratory tests include culture and antigen detection from respiratory samples, and detection of serum *Aspergillus* antigen and antibodies.

Detection of the galactomannan antigen (a polysaccharide that is released by *Aspergillus species*) in bronchoalveolar (BAL) samples is a useful and reliable diagnostic antigen test for the screening and management of aspergillosis [3]. However, BAL is obtained invasively and may not be feasible in critically ill patients and those on anticoagulants. Sputum could be an alternative sample to BAL in galactomannan antigen testing. In clinical laboratories, sputum is the commonest respiratory sample sent for microbiological investigations, since it is easily collected and lung specific. To our knowledge, only three studies have attempted to evaluate sputum GM detection within a very limited patient population group. Kimura et al. evaluated sputum GM in neutropenic patients whereas Baxter et al. looked at cystic fibrosis patients and reported a cut-off value of 1.22 and 0.5, respectively [4,5]. Both these researchers suggested that sputum GM detection is useful in the diagnosis of aspergillosis. In contrast, study on sputum GM utility in the diagnosis of pulmonary aspergillosis by Fayemiwo et al. was inconclusive [6].

In agreement with Kimura et al. and Baxter et al., we hypothesised that sputum could be useful in the diagnosis of pulmonary aspergillosis. Although the work of the above researchers supports sputum GM utility in the diagnosis of pulmonary aspergillosis, it was conducted in a limited patient population and may not be applicable to other patient populations. Therefore, the aim of this study was to retrospectively validate sputum GM diagnostic utility in pulmonary aspergillosis (chronic pulmonary aspergillosis and allergic bronchopulmonary aspergillosis) in a cohort of patients with chronic respiratory diseases.

## 2. Material and Methods

### 2.1. Study Population and Patient Selection

We retrospectively reviewed routinely collected sputum for galactomannan (GM) analysis from patients referred to Royal Brompton and Harefield Hospitals, a large tertiary cardiothoracic centre in London, between January 2015 and April 2019. To evaluate the concordance of sputum GM with culture and *Aspergillus* IgG, only patients with at least one sputum GM, culture, and *Aspergillus* serology performed within four weeks were included in the study.

Data were extracted from laboratory information systems (Winpath ClinSys 5.34) and an electronic patient record system (Servelec, UK). Baseline data for patients, including gender, age, clinical diagnosis (medical notes based), GM, *Aspergillus* IgG, and fungal culture, were collected.

### 2.2. Case Definitions, Patient Classification, and Reference Tests

Patients were stratified by fungal disease subtypes or underlying chronic lung disease according to international consensus criteria [2,7]. Colonisation was defined as individuals with ≥1 culture isolate of *Aspergillus species,* but without clinical or radiological evidence of pulmonary fungal disease. Three clinical categories were identified. These were chronic pulmonary aspergillosis (CPA), allergic bronchopulmonary aspergillosis (ABPA), and *Aspergillus* colonisation. The rest of the cohort, who was not diagnosed with pulmonary aspergillosis or *Aspergillus species* colonisation, was considered as a control group. There were 16 patients who were only *Aspergillus* IgG positive and had no clinical and radiological findings consistent with aspergillosis. This group was excluded from further analysis.

To evaluate the performance of sputum galactomannan (GM), two comparators were used. Firstly, a comparison with clinical diagnosis of CPA, ABPA, and *Aspergillus* colonisation (‘true positives’) and control group samples (‘true negatives’) was performed. Secondly, performance was compared to positive *Aspergillus* culture and *Aspergillus* IgG ELISA. Culture and *Aspergillus* IgG negative samples were considered as true negatives.

Optimal sputum GM index cut-off (CO) was estimated using the entire cohort via two approaches. In approach one, CO was obtained for combined CPA, ABPA, and colonisation, whereas in approach two, CO was assessed by using mycological evidence as indicated by either positive *Aspergillus* culture or *Aspergillus* IgG.

### 2.3. Microbiological Analysis

#### 2.3.1. Fungal Culture

Sputum sample was inoculated onto Sabouraud Dextrose Chloramphenicol agar (BIOMERIEUX, Culture media, Basingstoke, UK) and incubated at 37 °C for two days and at 30 °C for another five days, except sputum samples from Cystic Fibrosis (CF). CF cultures were incubated at 30 °C for four weeks. *Aspergillus* spp. were identified by culture characteristics supplemented with Matrix-Assisted Laser Desorption/Ionisation-Time of Flight (Bruker, MALDI-TOF, MS, Coventry, UK).

#### 2.3.2. Galactomannan

Galactomannan (GM) analysis on sputum was performed by sandwich enzyme-linked immunosorbent assay (PLATELIA^TM^
*ASPERGILLUS* Ag, BIO-RAD, Watford, UK) as per the manufacturer’s instruction. Sputum GM was processed as a part of routine mycological analysis when no bronchoalveolar fluid was available. 0.5 mL of expectorated sputum was homogenized with equal volume of phosphate buffered saline. Then, samples were processed using BAL GM analysis methodology. Moreover, sputum GM results were interpreted as per BAL methodology, but issued with comments indicating the lack of sputum GM validation by the kit manufacturer.

### 2.4. Aspergillus IgG

Serum samples were routinely tested for total specific *A fumigatus* immunoglobulin G using ImmunoCap assay (ImmunoCAP^TM^, Thermo Fisher, UK). Positive cut-off value of >40 mgA/L was used for all patients except for Cystic Fibrosis patients, whose cut-off was >90 mgA/L.

### 2.5. Data Analysis

Statistical analysis was performed using IBM SPSS statistics for Windows, version 27.0 (IBM Corp., Armonk, NY, USA). Categorical variables were summarised by using frequencies and percentages. Sputum galactomannan (GM) values difference of patient groups were compared using Mann-Whitney U test or Kruskal-Wallis test and *p* < 0.05 was considered significant. In addition, inter-rater reliability and Cohen’s Kappa were used to evaluate diagnostic performance and agreement of sputum GM with clinical categories, *Aspergillus* culture and *Aspergillus* IgG. Optimal cut-off point for sputum GM for the entire cohort was determined by receiver operating characteristics (ROC) curve and Younden’s index.

## 3. Results

In total, 218 patients met the inclusion criteria, and had sputum galactomannan (GM) and serum *Aspergillus* IgG tests (*Aspergillus* IgG) performed within four weeks. These comprised 102 males and 116 females with a median age of 58 years (46–69). The predominant clinical groups were chronic pulmonary aspergillosis (CPA) and allergic bronchopulmonary aspergillosis (ABPA), comprising 74% of the cohort (Table 1). Thirteen patients had *Aspergillus* colonisation; 27 (12%) patients did not have clinical or microbiological evidence of fungal disease and were considered disease controls. The predominant underlying diagnosis in these cases was idiopathic bronchiectasis (60%) and chronic obstructive pulmonary disease (23%); 16 (7%) patients had only *Aspergillus* IgG positivity and were excluded from analysis.

The study population had median sputum galactomannan (GM) and *Aspergillus* IgG index values of 1.25 (0.415–4.35) and 64 (35–103), respectively (Table 1). Among clinical groups, the CPA group had the highest GM index value, followed by the *Aspergillus* colonisation group. The *Aspergillus* IgG index value also showed a similar trend in this group (Table 1). The control group had a significantly lower median galactomannan index value compared with clinical groups (*p* < 0.001) (Figure 1). This was also seen when control group was compared to individual disease groups including CPA (*p* < 0.001), ABPA (*p* = 0.018) and colonisation (*p* = 0.002) (Figure 2).

### 3.1. Evaluating Sputum Galactomannan against Clinical Diagnostic Groups

Overall, sputum galactomannan (GM) showed fair agreement with clinical diagnostic groups with Kappa value of 0.3 (95% CI: 0.2, 0.5; *p* < 0.001) and sensitivity and specificity of 79% and 63%, respectively (Table 2). Among clinical groups, sputum GM showed the highest concordance in CPA group with moderate Kappa value of 0.5 (95% CI: 0.3, 0.7; *p* < 0.001). In the CPA group, sputum GM sensitivity was 86% and specificity was 63% (Table 2). In contrast, sputum agreement with ABPA was low with Kappa value of 0.1.

### 3.2. Evaluating Sputum Galactomannan Agreement with Mycological Evidence of Aspergillus Species

Mycological evidence of *Aspergillus species* was defined as the presence of positive culture or *Aspergillus* IgG or both. Patients with positive *Aspergillus* culture or *Aspergillus* IgG had significantly higher median sputum GM index content (*p* < 0.001) (Figure 3). When combined culture and *Aspergillus* IgG was used as comparator, the overall agreement of sputum GM with mycological evidence was moderate, giving a Kappa value of 0.5 (95% CI: 0.3, 0.6; *p* < 0.001) (Table 3). Concordance of sputum GM with *Aspergillus* IgG was better compared with culture.

Generally, sputum GM specificity was lower than sensitivity in this comparison. This difference between specificity and sensitivity was most striking in the CPA group when sputum GM was evaluated against culture. In this group, sputum GM had specificity of 22% and sensitivity of 100% when compared with culture (Table 3). Furthermore, sputum GM showed lower specificity against culture compared with *Aspergillus* IgG. For instance, the overall specificity of sputum GM was 38% against culture and 56% against *Aspergillus* IgG.

### 3.3. Evaluating Optimal Cut-Off of Sputum Galactomannan

We further estimated overall sputum GM cut-off (CO) of the entire cohort using two methods: analysing against clinical definitions of fungal disease and a disease control group as defined above, and using mycological status based on culture and *Aspergillus* IgG. When analysing against clinical diagnosis, the receiver operating characteristics (ROC) analysis gave an optimal GMI cut-off of 0.71. At a cut-off of 0.71, the sensitivity and specificity were 70% and 71%, respectively, with a Youden’s J index of 0.4 (AUC 0.74; CI: 0.65, 0.83; *p* < 0.001) (Figure 4).

Receiver operating characteristic (ROC) analysis against mycological evidence gave an optimal GMI cut-off of 0.71. At a cut-off of 0.71, the sensitivity and specificity were 77% and 78%, respectively, with a Youden’s J index of 0.6 (AUC 0.79; C I: 0.71, 0.86; *p* < 0.001) (Figure 5).

## 4. Discussion

Sputum is non-invasive, easily collected, and a lung-specific sample. In recognition of these advantages, we attempted to evaluate the utility of sputum galactomannan (GM) in the diagnosis of pulmonary aspergillosis. We report retrospective clinical validation of sputum GM against a chronic lung disease cohort including chronic pulmonary aspergillosis (CPA), allergic bronchopulmonary aspergillosis (ABPA), *Aspergillus* colonisation, and disease control groups (idiopathic bronchiectasis, cystic fibrosis (CF), chronic obstructive pulmonary disease (COPD), asthma, and interstitial lung disease (ILD)). For the first time, in this study, we show that sputum GM has satisfactory agreement with both clinical diagnosis of pulmonary fungal disease and mycological evidence comparators, with acceptable sensitivity and specificity.

The study population has a median sputum GM index value of 1.25 (0.415–4.35). Among clinical groups, CPA group had the highest median GM index value of 2.55 (0.920–5.84). The ABPA and colonisation groups had median sputum GM index values of 1.49 (0.492–3.98) and 1.89 (0.734–5.78). Of note, the colonization group had the highest sputum GM positivity rate (92%). This is probably due to higher organism load since all patients were Aspergillus species culture positive. The control group had a median GM index value of 0.378 (0.160–1.15), which was significantly lower compared with clinical groups (*p* < 0.001). These average values are close to those reported by other researchers. For example, Baxter et al. reported a sputum mean GM index value of 2.96 ± 2.11 for patients with *Aspergillus* airway infection, whereas Fayemiwo et al. reported a median sputum GM index value of 1.83 for the ABPA group [5,6]. In addition, Fayemiwo et al. reported a higher average GMI index in CPA compared with ABPA, concurring with our finding. Although the GMI difference between the CPA and ABPA groups was not statistically significant in our study, the observed higher average value in this finding and in previous studies may be indicative of high infection and organism load in the CPA group.

The overall agreement of sputum GM with clinical diagnosis was fair, whilst agreement with mycological evidence was moderate. Sputum GM concordance with clinical diagnosis gave an overall Kappa value of 0.3 (95% CI: 0.2, 0.5; *p* <0.001) and sensitivity and specificity were 79% and 63%, respectively. Among clinical groups, sputum GM showed relatively high agreement with the CPA group with a Kappa value of 0.5 (95% CI: 0.3, 0.7; *p* < 0.001), whilst agreement was poor with the ABPA group (Table 2). This poor concordance could be explained by low sputum GM positivity in the ABPA group (Table 1) compared with other clinical groups, which may be indicative of low organism load in this group.

In comparison with mycological evidence, sputum GM overall concordance improved to moderate (Table 3). Specificity was 70% and sensitivity was 85% with a Kappa value of 0.5 (95% CI: 0.3, 0.6; *p* < 0.001). In this comparison, sputum GM showed the highest concordance with mycological evidence in the CPA group, giving a Kappa value of 0.6 (0.3, 0.9; *p* < 0.001) and specificity and sensitivity of 71% and 94%, respectively.

The above reported overall specificity and sensitivity of sputum GM are close to those reported by Kimura et al. [4]. Kimura et al. compared sputum GM with bronchoalveolar GM and reported a sensitivity of 100% and specificity of 63% in haematological patients. This high sputum GM sensitivity in our study and the previous study could make this test more suitable for aspergillosis screening, as previously suggested by Kimura et al.

Sputum GM sensitivity was generally much higher than the specificity in all comparisons in this study. The difference between specificity and sensitivity was most evident when sputum GM was evaluated against culture in the CPA group. CPA group sensitivity against culture was 100% whereas specificity was 22%. Lower specificity of sputum GM, especially against culture, might be explained by the nature and limitation of the control used in this study and inherently low culture sensitivity. Almost 40% of the control group patients were GM positive, which should not be surprising as this group included patients with respiratory diseases such as COPD that is associated with *Aspergillus* colonization [8]. This apparent false positive depressed the specificity of sputum GM. Similarly, inherently low culture sensitivity has effect on the calculated specificity of sputum GM. In this cohort, only 30% of the cohort was culture positive (Table 1), predominantly growing *Aspergillus fumigatus*. This low growth percentage is indicative of the low culture sensitivity and consistent with previous studies [8,9].

In our study, we have determined sputum GM optimal cut-off (CO) value of 0.71 when the receiver operating characteristic curve (ROC) was constructed by using clinical aspergillosis groups and the control group. At this CO, sensitivity and specificity were 71% and 70%, respectively (Figure 3). Of note, the same GM CO value was found when the ROC was constructed using the presence and absence of *Aspergillus species* mycological evidence as the reference test. However, this method gave slightly higher sensitivity and specificity percentage. In this setting, sensitivity of 77% and specificity of 78% were found (Figure 4). The above CO is within the range of cut-off values reported for sputum GM by Baxter et al. and Kimura et al. This CO value is also close to the CO reported for bronchoalveolar lavage galactomannan by Wu et al. [10]. At the above CO, sensitivity and specificity were 77% and 78%, respectively with Younden’s index of 0.6 when *Aspergillus species* presence was used as a reference test.

Despite the limitations of our study, the above findings, especially the high sensitivity shown by the sputum galactomannan test, have clinical implications. Sputum GM had satisfactory concordance with both clinical diagnosis and mycological evidence. It also gave acceptable sensitivity and specificity when evaluated against clinical diagnosis and mycological evidence. With a cut-off value of 0.71 and sensitivity of 77% and specificity of 78%, the sputum galactomannan antigen test could be an adjunct to the *Aspergillus* antibody test in the diagnosis of pulmonary aspergillosis. The high sensitivity shown in this study may also make this non-invasive antigen test suitable for pulmonary aspergillosis screening.

## 5. Conclusions

This study showed that sputum galactomannan test has utility in the diagnosis of chronic pulmonary aspergillosis and allergic bronchopulmonary aspergillosis. Sputum GM showed satisfactory concordance with the clinical diagnosis of aspergillosis, *Aspergillus* culture, and *Aspergillus* IgG. An optimal cut-off of 0.71 was determined. At the cut-off point of 0.71, sputum GM had a sensitivity and specificity of 77% and 78%, respectively with a Youden index (J) of 0.6 (AUC: 0.83; CI: 0.69–86, *p* < 0.001). A prospective study in a cohort of well-characterised patients is required to further validate this potentially useful test.

## 6. Limitations

This study has a number of limitations, of which the most important ones are being retrospective, single-centre and the lack of a proper negative control group. Our control group contained patient with respiratory diseases that are associated with *Aspergillus* colonisation such as chronic obstructive diseases (Guinea et al., 2010). However, this control group had a significantly lower median GM index value, which was below the positive cut-off value. We tried to reduce the impact of the above limitation by using *Aspergillus species* mycological evidence as a reference test in the entire cohort in addition to clinical diagnosis. In addition, data on whether patients were exposed to antifungals prior to sputum collection were not available. Administration of these agents could have an impact on both galactomannan and Aspergillus IgG index values.

## Figures and Tables

**Figure 1 jof-08-00188-f001:**
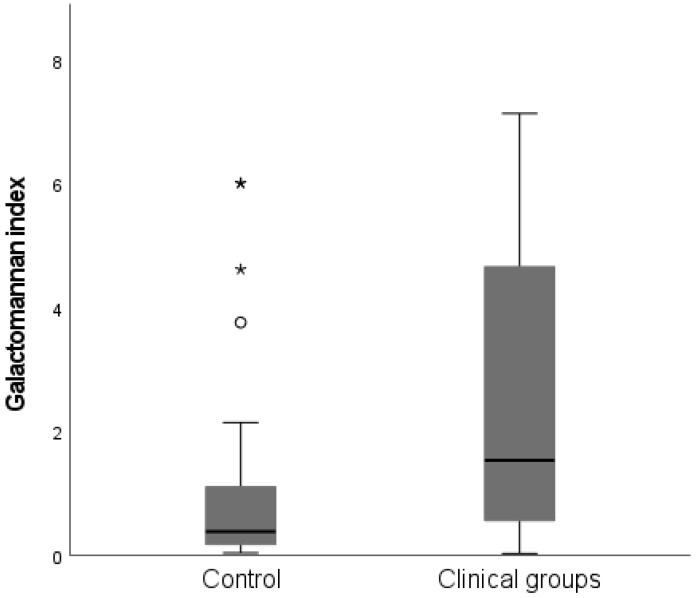
Galactomannan index (GMI) values box plot for control group and combined clinical groups (CPA, ABPA and colonisation). Control group has significantly lower median GMI compared with the clinical groups (*p* < 0.001).

**Figure 2 jof-08-00188-f002:**
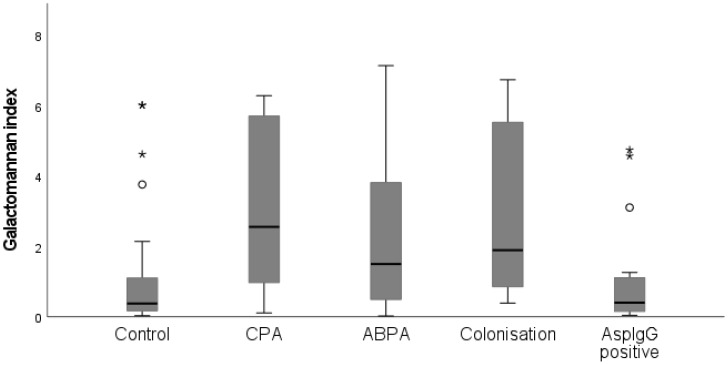
Galactomannan index (GMI) values box plot for control group, chronic pulmonary aspergillosis (CPA), allergic bronchopulmonary aspergillosis (ABPA), colonization, and Aspergillus IgG positive (AspIgG positive). Medians GMI of the control and the other groups were compared using Kruskal-Wallis test. Control group had significantly lower median GMI value compared with CPA (*p* < 0.001), ABPA (*p* = 0.018), and colonisation (*p* = 0.002).

**Figure 3 jof-08-00188-f003:**
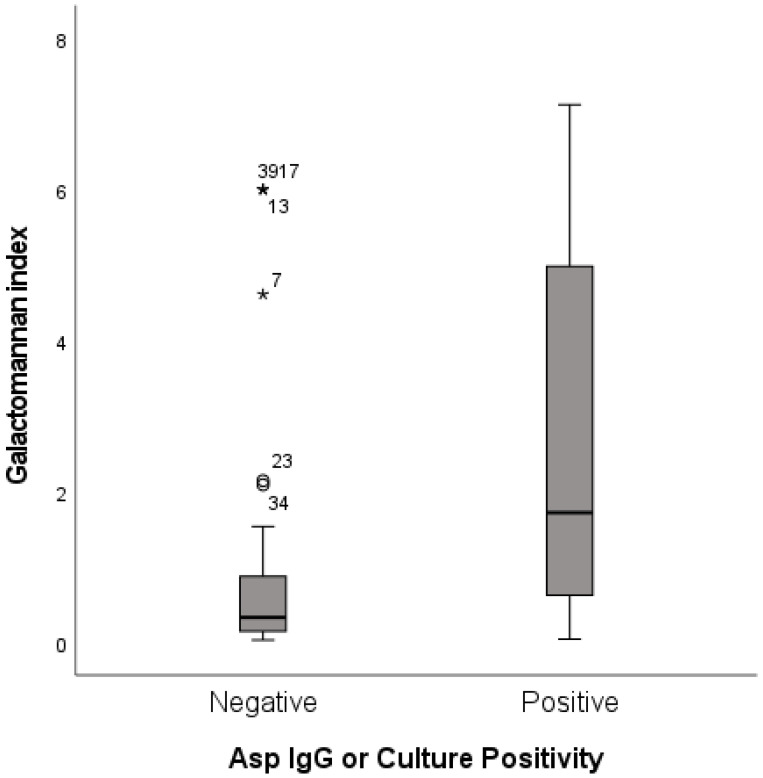
Galactomannan index (GMI) values box plot for Aspergillus IgG/culture positive and negative groups. Groups were compared by using Mann-Whitney U test. Control group had significantly lower GMI value compared with Aspergillus IgG or culture positive group. (*p* < 0.001).

**Figure 4 jof-08-00188-f004:**
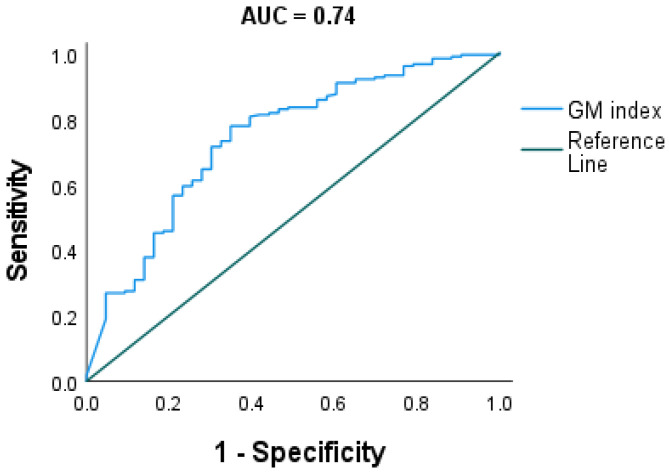
Receiver operating characteristic curve (ROC) for sputum galactomannan index value using Aspergillosis clinical diagnosis as comparator. The area under the curve (AUC) was 0.74 (95% CI: 0.65–0.83, *p* < 0.001). Optimal cut-off (CO) was 0.71 with Younden’s J index of 0.4. At this CO, sensitivity and specificity were 71% and 70%, respectively.

**Figure 5 jof-08-00188-f005:**
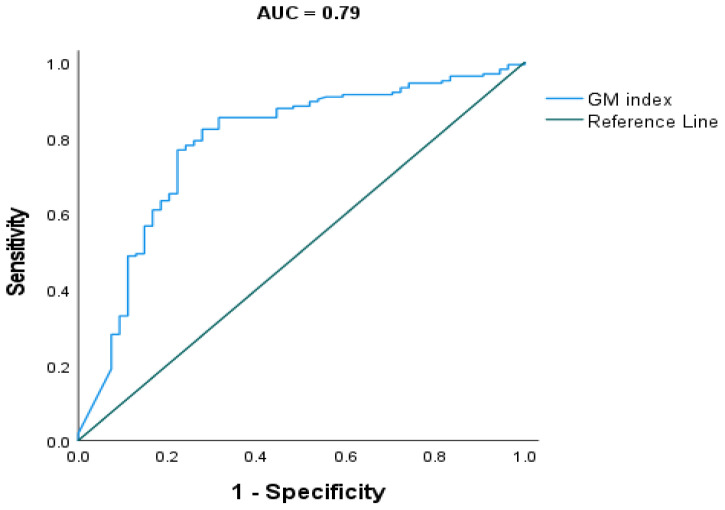
Receiver operating characteristic curve (ROC) for sputum galactomannan index value using mycological evidence as comparator. The area under the curve (AUC) was 0.79 (95% CI: 0.0.71–0.86, *p* < 0.001). Optimal cut-off (CO) was 0.71 with Younden’s J index of 0.6. At this CO point, sensitivity and specificity were 77% and 78%, respectively.

**Table 1 jof-08-00188-t001:** Demographics of the study population (*n* = 218) and descriptive statistics of chronic pulmonary aspergillosis, allergic bronchopulmonary aspergillosis, colonization, and control group.

	CPA	ABPA	Colonisation	Control Group	*Aspergillus* IgG Positive	Total
Number (%)	57 (26)	105 (48)	13 (6)	27 (12)	16 (7)	218 (100)
Age (IQC)	55 (44–63)	61 (49–71)	62 (34–73)	54 (34–65)	58 (10–67)	58 (46–69)
Sex (Female/Male)	34/23	50/55	3/13	15/13	8/8	116/102
GM positive	86%	74%	92%	40%	44%	72%
GMI, median (IQC)	2.55 (0.92–5.84)	1.49 (0.492–3.98)	1.89 (0.734–5.78)	0.378 (0.160–1.15)	0.399 (0.140–1.18)	1.25 (0.415–4.35)
*Aspergillus* IgG positive	82%	72%	69%	37%	100%	69%
*Aspergillus* IgG index, median (IQC)	99 (57–150)	63 (36–100)	81 (43–109)	32 (20–38)	60 (50–88)	64 (35–103)
Culture	35%	31%	100%	None	None	30%

CPA: chronic pulmonary aspergillosis; ABPA: allergic bronchopulmonary aspergillosis; GM: galactomannan; GMI: galactomannan index; IQC: interquartile range.

**Table 2 jof-08-00188-t002:** Sputum galactomannan agreement with clinical diagnosis; specificity, sensitivity, and Kappa values with 95% confidence interval (CI) are shown.

	All Patient	CPA	ABPA	Colonisation
Specificity (95% CI)	63 (43, 80)	63 (43, 80)	63 (43, 80)	63 (43, 80)
Sensitivity (95% CI)	79 (73, 85)	86 (72, 92)	74 (65, 82)	92 (62, 100)
Kappa value (95% CI)	0.3 (0.2, 0.5)	0.5 (0.3, 0.7)	0.1 (0.06, 0.1)	0.2 (0.1, 0.3)
*p* value	<0.001	<0.001	<0.001	0.001

**Table 3 jof-08-00188-t003:** Sputum galactomannan agreement with *Aspergillus* culture, *Aspergillus* IgG, and *Aspergillus* culture/*Aspergillus IgG;* specificity, sensitivity, and Kappa values with 95% confidence interval (CI) are shown.

	*Aspergillus* Culture			*Aspergillus* IgG			*Aspergillus* Culture/IgG		
Clinical Entity	All Patients	CPA	ABPA	All Patients	CPA	ABPA	All Patients	CPA	ABPA
Specificity (95% CI)	38 (30, 46)	22 (10, 39)	33 (23, 41)	56 (44, 68)	60 (27, 86)	55 (36, 73)	70 (56, 82)	71 (30, 95)	79 (54, 93)
Sensitivity (95% CI)	94 (84, 98)	100 (80, 100)	91 (75, 98)	85 (78, 90)	96 (84, 99)	86 (75, 92)	85 (79, 90)	94 (82, 98)	86 (77, 92)
Kappa Value (95% CI)	0.2 (0.15, 0.30)	0.2 (0.01, 0.3)	0.2 (0.1, 0.3)	0.4 (0.3, 0.6)	0.6 (0.3, 0.9)	0.4 (0.2, 0.6)	0.5 (0.3, 0.6)	0.6 (0.3–0.9)	0.6 (0.4, 0.8)
*p* value	<0.001	0.025	0.008	<0.001	<0.001	<0.001	<0.001	<0.001	<0.001

## Data Availability

This study didn’t report any data; anonymized data is stored secure NHS trust server.

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
