# Peer review of "Sputum Galactomannan Has Utility in the Diagnosis of Chronic Pulmonary Aspergillosis"

_jof, 2022, doi:10.3390/jof8020188_

Round 1

Reviewer 1 Report

The article is interesting and provide systematic analysis of GM test for assist diagnosis of CPA. A few minor comments:

  1. Page 3: Please explain the sample processing of the sputum for GM test. Did you do any centrifugation? Did you do sputum induction?
  2. Page 2, last paragraph.

“The rest of the cohort, where there was no clinical or mycological diagnosis of pulmonary aspergillosis, were considered as a control group. There were 16 patients who were only Aspergillus IgG positive. This group was excluded from further analysis.”

In table 1 the control group showed the positivity rate of 37% of Asp IgG. Do you mean the mycological diagnosis in the text is Asp culture?  

  1. The GM positivity rate in Table 1 is 92%, the highest among all groups. Could it be because of cross-reaction with other fungal species or the patients had cavity or bullae in the radiographs that might predispose Asp colonisation? Please discuss this also in the Discussion.
  2. What type of Asp species recovered from the culture? Is it dominated by fumigatus?
  3. Page 9 and 10. Please add information in Figure 4 and 5 what type of CO analysis used, which type of population etc.
  4. Page 9. Could you please describe the group of patients used in CO analysis? Is the CO produced using CPA (57 patients) vs control group (27 patients) only?

Author Response

POINT BY POINT RESPONSE TO THE REVIEWS’ COMMENTS

Response to reviewer 1.

We thank the reviewer for reviewing our manuscript and for the detailed comments and suggestions.  We have revised the manuscript in line with these comments and suggestions. 

  1. Page 3: Please explain the sample processing of the sputum for GM test. Did you do any centrifugation? Did you do sputum induction?

Response: explained and clarified.  An aliquot of expectorated sputum was homogenised with equal volume of phosphate buffered saline.  This sample was processed as per bronchoalveolar lavage sample according manufacturer’s instruction. 

  1. Page 2, last paragraph.

“The rest of the cohort, where there was no clinical or mycological diagnosis of pulmonary aspergillosis, were considered as a control group. There were 16 patients who were only Aspergillus IgG positive. This group was excluded from further analysis.”

In table 1 the control group showed the positivity rate of 37% of Asp IgG. Do you mean the mycological diagnosis in the text is Asp culture?  

Response: clarified; here mycological diagnosis we have meant Aspergillus culture positive patients, who could be classified as colonisation according to consensus criteria.  16 patients who were Aspergillus IgG positive and had no clinical aspergillosis diagnosis didn’t fit into this consensus.  We also could use them as control since Aspergillus IgG test is gold standard for the diagnosis of Aspergillosis. 

  1. The GM positivity rate in Table 1 is 92%, the highest among all groups. Could it be because of cross-reaction with other fungal species or the patients had cavity or bullae in the radiographs that might predispose Asp colonisation? Please discuss this also in the Discussion.

Response: thanks, you for pointing this out.   The cohort we assessed almost all had some sort of respiratory disease such COPD that may predispose them to Aspergillus infection or colonisation.  Colonisation had highest GM positive rate, which may be due constant release of the antigen by growing fungi- they were all culture positive. 

  1. What type of Asp species recovered from the culture? Is it dominated by fumigatus?

Response: A fumigatus was predominant organism in the culture, almost 90%. 

Briefly mention this in the discussion

  1. Page 9 and 10. Please add information in Figure 4 and 5 what type of CO analysis used, which type of population etc.

Response: information added many thanks. 

  1. Page 9. Could you please describe the group of patients used in CO analysis? Is the CO produced using CPA (57 patients) vs control group (27 patients) only?

 Response: Here comparators were a) presence and absence of clinical diagnosis of aspergillosis and b) presence and absence of mycological evidence as indicated by either positive Aspergillus culture or Aspergillus IgG (or both). 

Thanks for pointing out the low number of control group.  It would have been more useful to use larger control group, however, were constrained by the data available and rigorous definition of control group.  For instance, we could have included 16 patients who were Aspergillus IgG but lacked clinical diagnosis of aspergillosis.  However, we didn’t use this group as control since Aspergillus IgG is gold standard tests for aspergillosis.  we have mentioned the small size of the control group in the limitation section. 

Reviewer 2 Report

This is a good article with an appropriate sample size to support results. Just some comments:

  1. Introduction: The authors need to bring clarity in the introduction that they are evaluating GM for the diagnosis of chronic and allergic aspergillosis, currently it is not that clear.
  2. Methods: The authors have mentioned that this is a retrospective study so it is not clear whether GM was performed on stored sputum samples or simultaneously with culture and other mycological specimens. If yes then whether serum GM is routinely performed at their center.
  3. Results: the patient who were excluded with only IgG positivity, did they also have clinical and radiological findings concurrent with CPA or ABPA? If yes then they should not be excluded
  4. Table 1: Please write all abbreviations in footnote
  5. Table 2 and 3 title: Kapp values should be Kappa values
  6. Language could be improved further 

Author Response

Response to reviewer 2

We thank the reviewer for reviewing our manuscript and for the detailed comments and suggestions.  We have revised the manuscript in line with these comments and suggestions. 

1. Introduction: The authors need to bring clarity in the introduction that they are evaluating GM for the diagnosis of chronic and allergic aspergillosis, currently it is not that clear.

Response: clarified in the aim of the study, thanks for the suggestion. 

2. Methods: The authors have mentioned that this is a retrospective study, so it is not clear whether GM was performed on stored sputum samples or simultaneously with culture and other mycological specimens. If yes then whether serum GM is routinely performed at their centre.

Response: clarified; expectorated sputum was processed for galactomannan analysis, when no bronchoalveolar fluid was available.  Culture and GM was processed simultaneously.  Thanks for pointing this out and our results were issued with warning that sputum is not validated by the manufacturer for GM analysis. 

3. Results: the patient who were excluded with only IgG positivity, did they also have clinical and radiological findings concurrent with CPA or ABPA? If yes then they should not be excluded

Response: clarified; this group didn’t have clinical and radiological finding consistent with CPA or ABPA and could not fit into Aspergillosis disease stratification. 

4. Table 1: Please write all abbreviations in footnote

Response: Footnote added

5. Table 2 and 3 title: Kapp values should be Kappa values

Response:  spelling corrected

6. Language could be improved further 

Response: tried and taken the suggestion onboard. Many thanks